# Towards a Geometric Theory of Fairness: Detecting Mode Collapse on the Grassmannian Manifold

**Beatriz Cardoso Nascimento & Marcos M. Raimundo**
Institute of Computing
University of Campinas (UNICAMP)
Campinas, SP, Brazil
`b247403@dac.unicamp.br, mraimundo@ic.unicamp.br`

## Abstract

Generative models frequently exhibit mode collapse, disproportionately failing on minority subpopulations. This phenomenon is central to fair representation learning. However, detecting these failures without ground-truth labels remains an open challenge, as standard Euclidean metrics are often dominated by high-dimensional stochastic noise. In this work, we propose a geometric perspective on this problem: we hypothesize that systematic "unfairness" manifests not as magnitude errors, but as stable, low-rank subspaces in the residual field, distinct from random noise. We introduce a diagnostic framework that lifts residuals to the Grassmann manifold, allowing us to analyze the "shape" of model failures. Providing proof-of-concept evidence on MNIST, we demonstrate that our Grassmannian metric successfully isolates the structural failure modes. These preliminary results suggest that geometry-grounded tools are promising for the next generation of blind fairness auditing.

## 1 Introduction

Many failures in generative modeling can be traced to a lack of coverage (Lala et al., 2018; Zhong et al., 2019). While GANs are known to drop modes (Richardson & Weiss, 2018), even likelihood-based models can effectively ignore minority structures to improve average fit, treating rare features as noise. A structurally identical issue arises in fair representation learning, where models minimize average reconstruction error at the expense of specific subpopulations (Pelegrina & Duarte, 2023).

Current approaches largely treat these failures as disparities in scalar metrics (e.g., difference in MSE) (Martinez et al., 2020). However, in high-dimensional spaces, scalar magnitudes are often uninformative: a systematic failure on a minority group and a random stochastic outlier may share the same Euclidean error norm. While recent work leverages local geometric descriptors to characterize the data manifold (Humayun et al., 2024), or topological (Khrulkov & Oseledets, 2018) and Grassmannian (Aboussalah & Abbahaddou, 2024) properties to quantify general coverage failures, we argue that standard fairness metrics remain geometrically blind to the root cause of bias: the systematic exclusion of data subspaces.

In this work, we propose a geometric perspective on fairness, framing it as a problem of *subspace stability*. We hypothesize that unlike stochastic noise — which is geometrically diffuse — systematic unfairness manifests as a *geometric mode collapse*, where the semantic structure of a group is relegated to a coherent, low-dimensional subspace within the residuals.

To explore this, we introduce an unsupervised diagnostic framework that lifts residual vectors to the Grassmann manifold (Hamm & Lee, 2008). Distinct from *neural collapse* in supervised learning (Xu et al., 2024) and subspace ensemble methods designed to enhance high-dimensional clustering accuracy (He et al., 2025), our framework leverages Grassmannian geometry to specifically diagnose structural fairness failures. In this preliminary work, we instantiate our framework using Factor Analysis (FA) due to its explicit noise model, which provides a tractable analytical baseline to validate our geometric hypotheses before extending the approach to non-linear generative models. We

present proof-of-concept experiments on MNIST data, demonstrating that while Euclidean baselines fail to distinguish bias from noise, our Riemannian approach successfully recovers the geometry of the failure modes. This suggests that geometry-grounded tools are promising for the next generation of blind fairness auditing.

## 2 METHODOLOGY

We propose a geometric framework to audit fairness. Our core premise is that unfairness manifests as a *structural collapse*, distinguishable from random noise.

### 2.1 DIAGNOSTIC METRIC: STRUCTURAL DEVIATION

To validate whether errors are structural (systematic) or stochastic (random), we require a statistical test. As a primary instantiation of our framework, we employ Factor Analysis as a generative model specifically because it offers an explicit noise model, unlike deep generative approaches, where noise is often implicit. While the geometric lifting process is agnostic to the underlying generator, FA provides a clear baseline for structural diagnostics, see Appendix C for details. Following the probabilistic formulation in Bishop & Nasrabadi (2006), the generative process is defined as $x = Wz + \mu + \epsilon$, where the noise $\epsilon$ is defined by a diagonal covariance matrix $\Psi$.

Consequently, in a well-calibrated model, the empirical residuals $r$ must converge to this theoretical distribution $r \sim \mathcal{N}(0, \Psi)$. This explicit baseline allows us to test for structural deviations. We quantify this using the Kullback-Leibler (KL) divergence between the empirical residual covariance and the model's theoretical noise $\Psi$.

A low $D_{KL}$ implies residuals are consistent with sensor noise. A high $D_{KL}$ indicates that the latent factors failed to capture significant variance, "leaking" structure into the residuals. This metric serves as a structural diagnostic: while Grassmannian lifting isolates candidate subspaces of failure, $D_{KL}$ confirms whether a given cluster represents a structural "leakage" of information or remains consistent with stochastic noise.

### 2.2 GEOMETRIC ERROR MODES AND LIFTING

Given that the residuals of neglected groups concentrate in specific directions, we formalize this failure as a geometric object, adapting the concept of *Temporal Modes* from Aboussalah & Abbahaddou (2024).

Let $R \in \mathbb{R}^{N \times D}$ be the matrix of residual vectors for a subset of samples (where each row $r_i$ is a residual). The **Geometric Error Mode** of rank $m$, denoted $\mathcal{M}_m(R)$, is defined as the $m$-dimensional principal subspace spanned by the top-$m$ right singular vectors of $R$. Formally, $\mathcal{M}_m(R) \in \text{Gr}(m, D)$, representing the dominant "shape" of the reconstruction failure.

To detect these modes in an unsupervised setting, we perform **Local Residual Lifting**. We target the subset $\mathcal{X}_{worst}$ of samples with the highest reconstruction error. For each sample $x_i \in \mathcal{X}_{worst}$, we estimate its local error geometry via three steps:

1. **Neighborhood:** Identify the indices of the $k$-nearest neighbors of $x_i$ in the original data space, denoting their set as $\mathcal{N}_k(i)$.

2. **Local Residual Matrix:** Construct the local residual matrix $E_i \in R^{k \times D}$ by stacking the residual vectors of these neighbors:

$$E_i = \begin{bmatrix} r_{n_1} \\ \vdots \\ r_{n_k} \end{bmatrix}, \quad \forall n_j \in \mathcal{N}_k(i) \tag{1}$$

3. **Subspace Estimation:** Apply Principal Component Analysis (PCA) to $E_i$. The lifted representation $\mathcal{S}_i \in \text{Gr}(m, D)$ is the subspace spanned by the top-$m$ principal components (orthonormal basis $Y_i \in \mathbb{R}^{D \times m}$):

$$\mathcal{S}_i = \text{span}(Y_i) \tag{2}$$

Intuitively, $\mathcal{S}_i$ represents the "tangent space of the failure" at $\boldsymbol{x}_i$. If the error is stochastic noise, $\mathcal{S}_i$ is unstable and random; if it is a mode collapse, $\mathcal{S}_i$ aligns consistently with the missing latent features of the minority group.

## 2.3 GRASSMANNIAN ANALYSIS

To cluster these local subspaces, we require a metric that is invariant to basis rotation. We employ the Projection Metric $d_P$ (Edelman et al., 1998; Wang et al., 2006), defined by the principal angles $\boldsymbol{\theta}$ between subspaces. For computational efficiency, we utilize the Projection Kernel (Hamm & Lee, 2008):

$$k(\mathcal{S}_i, \mathcal{S}_j) = ||\boldsymbol{Y}_i^\top \boldsymbol{Y}_j||_F^2 = \sum_{l=1}^m \cos^2 \theta_l \tag{3}$$

where $\boldsymbol{Y}_i, \boldsymbol{Y}_j$ are the orthonormal bases defined in Eq. 2. Maximizing this kernel is equivalent to minimizing the squared projection distance, as $d_P^2(\mathcal{S}_i, \mathcal{S}_j) = m - k(\mathcal{S}_i, \mathcal{S}_j)$. By applying Spectral Clustering on the kernel matrix $\boldsymbol{K}$, we identify "geometric modes" of unfairness—clusters of samples sharing the same structural failure—without requiring explicit geodesic optimization.

## 3 EXPERIMENTS

Theoretical results suggest that the separability of signal classes is determined by the principal angles between their subspaces (Huang et al., 2015). We leverage this to treat "fairness failure modes" as geometric classes. We validate our framework on the MNIST dataset to demonstrate applicability to high-dimensional visual data.

**Setup.** To evaluate the framework on visual data, we constructed an unfair scenario using MNIST digits '1' (majority, $N = 3000$) and '7' (minority, $N = 30$). We trained a Factor Analysis model ($z = 15$) on this imbalanced set. The model collapsed on the minority class, yielding a reconstruction error significantly higher than the majority (MSE 52.3 vs. 6.6).

**Methodology.** We analyzed the residuals of the test data using our Grassmannian clustering method with $K = 10$. Our goal was to determine if the algorithm could automatically group the failing samples and if the KL divergence could distinguish structural collapse from standard noise without access to ground-truth labels. We compared this against a *Balanced Scenario* ($N = 2000$ per class).

**Results: Detection via KL Disparity.** As detailed in Table 1, the proposed method successfully isolated the collapsed samples into distinct clusters and flagged them via the KL metric.

- **Control Scenario:** In the balanced model (Table 1, Left), the clustering grouped samples by digit type (e.g., Cluster 3 and 9 are 100% minority). Crucially, the KL divergence remained stable across all groups ($D_{KL} \approx 2.0 \times 10^3$ to $3.1 \times 10^3$). This confirms that in a working model, the residual geometry of the minority class is statistically similar to the majority.

- **Biased Scenario:** In the collapsed model (Right), the algorithm concentrated the minority samples into specific clusters (IDs 5, 9, 1, 6). The KL divergence for these groups exploded by orders of magnitude, reaching values as high as $2.7 \times 10^5$.

To understand the nature of the detected collapse, we visually inspected the samples within the highest-divergence clusters (Figure 1).

In the **Biased Scenario**, the cluster flagged with the most extreme KL divergence (278,259) is composed of digits exhibiting specific geometric traits: distinct horizontal strokes and acute angles. This supports the hypothesis that the biased model—trained predominantly on the digit '1'—learned a representation biased towards vertical features. Consequently, instances of the minority class '7' that are morphologically closest to a '1' yield lower structural errors, whereas samples with pronounced horizontal components (the "crooked" or standard handwritten '7') are penalized most severely.

Table 1: **Structural Diagnosis Results** ($K = 10$). Comparison of KL Divergence in local residual subspaces. **Left (Control):** Minority clusters maintain low KL divergence. **Right (Biased):** The method isolates minority samples into clusters (IDs 5, 9, 1, 6) with massive KL divergence, flagging structural collapse.

| | BALANCED CONTROL | | | BIASED MODEL | |
|---|---|---|---|---|---|
| **ID** | **% '7'** | **KL Div** | **ID** | **% '7'** | **KL Div** |
| 3 | 100.0% | 3174.2 | **5** | **100.0%** | **278,258.6** |
| 9 | 100.0% | 2482.7 | **9** | **100.0%** | **40,734.6** |
| 8 | 0.0% | 2455.5 | **1** | **100.0%** | **40,472.2** |
| 6 | 100.0% | 2360.6 | **6** | **100.0%** | **33,400.3** |
| 7 | 0.0% | 2331.2 | 8 | 0.0% | 2037.2 |
| 1 | 100.0% | 2227.6 | 7 | 0.0% | 1885.3 |
| 5 | 2.8% | 2207.2 | 4 | 0.0% | 1819.2 |
| 0 | 0.0% | 2161.5 | 0 | 2.8% | 1787.3 |
| 2 | 0.0% | 2155.5 | 2 | 0.0% | 1649.0 |
| 4 | 0.0% | 2078.2 | 3 | 0.0% | 1639.1 |

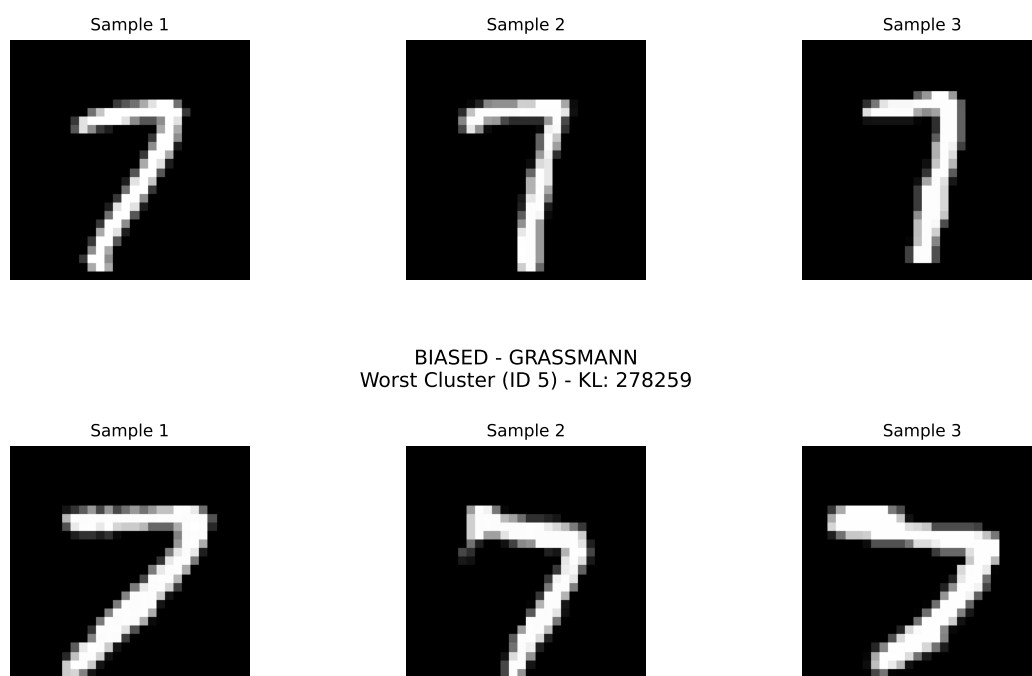

BALANCED - GRASSMANN
Worst Cluster (ID 3) - KL: 3174

Sample 1       Sample 2       Sample 3

BIASED - GRASSMANN
Worst Cluster (ID 5) - KL: 278259

Sample 1       Sample 2       Sample 3

Figure 1: **Qualitative Inspection of Detected Modes.**
**Top** (Balanced Scenario, KL $\approx 3.1 \times 10^3$): Representative samples from the worst-performing cluster in the fair model. The digits appear standard, and the low KL divergence indicates these residuals are essentially random noise.
**Bottom** (Biased Scenario, KL $\approx 2.7 \times 10^5$): Samples from the cluster with the highest structural error. Note the morphology of these digits: they exhibit a consistent angular deviation from the vertical majority class ('1'). Quantitatively, these failure modes align within a stable range of principal angles (averaging $\approx 63.5°$), confirming that the Grassmannian framework isolates a coherent geometric structure rather than diffuse stochastic noise.

## 4    CONCLUSION

In this work, we proposed a geometric perspective on fairness auditing, operating on the hypothesis that systematic failures manifest as stable, low-rank structures on the Grassmann manifold. Our proof-of-concept experiments on MNIST validated this premise: the proposed lifting framework successfully clustered and flagged minority samples based solely on their residual geometry, specifically isolating instances with morphological deviations orthogonal to the majority class. By characterizing the "shape" of these failures, we demonstrate that structural bias is topologically distinct from stochastic noise. These results identify an open problem at the intersection of geometry and learning: moving beyond scalar error magnitudes to leverage differential geometry for blind auditing, and ultimately, to enforce isometric coverage in the next generation of generative systems.

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

## A    NOTATION

To facilitate ease of reference and ensure mathematical clarity, Table 2 provides a comprehensive summary of the notations, definitions, and dimensional spaces for the key variables introduced throughout the methodology and experiments.

Table 2: Summary of Notation and Dimensions

| Symbol | Description | Dimensions / Space |
|--------|-------------|--------------------|
| $N$ | Number of samples in the analyzed subset $\mathcal{X}_{worst}$ | $\mathbb{Z}^+$ |
| $D$ | Dimensionality of the original data space (e.g., pixels) | $\mathbb{Z}^+$ |
| $k$ | Number of nearest neighbors for local lifting | $\mathbb{Z}^+$ |
| $m$ | Rank (dimension) of the geometric error mode | $\mathbb{Z}^+$ |
| $\boldsymbol{r}_i$ | Residual vector of sample $i$ | $\mathbb{R}^D$ |
| $\boldsymbol{R}$ | Global residual matrix for the analyzed subset | $\mathbb{R}^{N \times D}$ |
| $\boldsymbol{E}_i$ | Local residual matrix for the neighborhood $\mathcal{N}_k(i)$ | $\mathbb{R}^{k \times D}$ |
| $\boldsymbol{Y}_i$ | Orthonormal basis representing the subspace $\mathcal{S}_i$ | $\mathbb{R}^{D \times m}$ |
| $\mathcal{S}_i$ | Lifted representation of error (Grassmannian point) | $\mathrm{Gr}(m, D)$ |
| $\boldsymbol{\Psi}$ | Noise covariance matrix from Factor Analysis | $\mathbb{R}^{D \times D}$ |
| $D_{KL}$ | Kullback-Leibler divergence | $\mathbb{R}^+$ |

## B    ANALYSIS OF STRUCTURAL FAILURE MODES

The complete cluster distributions presented in Table 3 and Table 4 provide empirical evidence for our geometric hypothesis. We identify three critical phenomena that distinguish our Grassmannian diagnostic framework from standard baselines:

**1. Fragmentation vs. Coherence:** In the *Biased Scenario*, the Euclidean baseline exhibits significant fragmentation. While it identifies high-KL regions, these are confined to very small, isolated groups (e.g., Euclidean IDs 8, 3, and 2 contain only 5 to 8 samples each). In contrast, the Grassmannian approach consolidates the failure mode into larger, coherent clusters (e.g., Grassmann IDs 5 and 1 contain 63 and 75 samples, respectively). This suggests that Euclidean metrics treat structural collapse as stochastic outliers, whereas the Grassmannian manifold successfully captures the underlying low-rank collapse.

**2. Rotational Invariance and Purity:** A comparison between the Stiefel and Grassmannian manifolds highlights the necessity of subspace-only analysis. The Stiefel manifold, which is sensitive to specific basis orientations (rotations), yields "noisy" clusters where minority and majority samples are often mixed (e.g., Stiefel ID 4 at 73% purity). The Grassmannian lifting achieves near-perfect purity (100% minority) in its top 4 clusters, confirming that the signature of the failure mode is defined by the subspace span rather than a particular coordinate alignment.

**3. Safeguard Against False Positives:** The *Balanced Scenario* results serve as a crucial validation of our metric's specificity. Although the Grassmannian method still groups minority samples based on their shared digit geometry (e.g., IDs 3 and 9), the KL divergence remains consistently low ($D_{KL} \approx 2k$–$3k$). This is orders of magnitude lower than the $278k$ observed during actual model collapse. This result proves that our framework does not merely flag minority groups, but specifically detects when their representation has structurally diverged from the model's learned distribution.

Table 3: **Biased Scenario.** Comparison of all 10 clusters across Grassmannian (Ours), Stiefel, and Euclidean methods. Note how Grassmannian consolidation yields larger, purer clusters compared to Euclidean fragmentation.

| | Grassmann (Ours) | | | | Stiefel (Baseline) | | | | Euclidean (Baseline) | | |
|---|---|---|---|---|---|---|---|---|---|---|---|
| ID | Count | %'7' | KL | ID | Count | %'7' | KL | ID | Count | %'7' | KL |
| 5 | 63 | 100% | 278,258.6 | 4 | 41 | 73% | 237,347.0 | 8 | 5 | 100% | 315,345.2 |
| 9 | 38 | 100% | 40,734.6 | 2 | 34 | 91% | 235,580.5 | 3 | 8 | 100% | 139,274.4 |
| 1 | 75 | 100% | 40,472.2 | 1 | 35 | 88% | 33,610.1 | 2 | 8 | 100% | 135,668.9 |
| 6 | 23 | 100% | 33,400.3 | 8 | 31 | 35% | 31,492.5 | 9 | 28 | 100% | 134,925.1 |
| 8 | 22 | 0% | 2,037.2 | 6 | 39 | 59% | 29,351.4 | 7 | 34 | 100% | 85,678.3 |
| 7 | 13 | 0% | 1,885.3 | 5 | 51 | 25% | 23,748.8 | 4 | 29 | 100% | 61,732.9 |
| 4 | 59 | 0% | 1,819.2 | 9 | 30 | 100% | 22,839.4 | 0 | 30 | 100% | 22,933.9 |
| 0 | 36 | 2.8% | 1,787.3 | 3 | 57 | 40% | 11,499.9 | 1 | 46 | 100% | 13,259.8 |
| 2 | 32 | 0% | 1,649.0 | 0 | 37 | 21% | 3,218.9 | 6 | 209 | 4.3% | 5,447.6 |
| 3 | 39 | 0% | 1,639.1 | 7 | 45 | 0% | 1,954.9 | 5 | - | - | - |

Table 4: **Balanced Scenario.** In a fair model, although clusters may still isolate the minority class based on geometric shape (e.g., Grassmann IDs 3, 9, 6), the KL divergence remains consistently low ($\approx 2k$–$3k$) across all methods. This confirms that the residual subspaces are statistically consistent with noise, providing a robust safeguard against false positives in fairness auditing.

| | Grassmann (Ours) | | | | Stiefel (Baseline) | | | | Euclidean (Baseline) | | |
|---|---|---|---|---|---|---|---|---|---|---|---|
| ID | Count | %'7' | KL | ID | Count | %'7' | KL | ID | Count | %'7' | KL |
| 3 | 40 | 100.0% | 3,174.2 | 0 | 34 | 100.0% | 2,573.8 | 2 | 8 | 100.0% | 6,096.1 |
| 9 | 27 | 100.0% | 2,482.7 | 3 | 48 | 81.2% | 2,527.9 | 9 | 20 | 100.0% | 3,004.8 |
| 8 | 22 | 0.0% | 2,455.5 | 4 | 71 | 71.8% | 2,503.7 | 8 | 29 | 100.0% | 2,584.8 |
| 6 | 48 | 100.0% | 2,360.6 | 7 | 42 | 69.0% | 2,438.8 | 0 | 6 | 100.0% | 2,477.8 |
| 7 | 14 | 0.0% | 2,331.2 | 8 | 28 | 32.1% | 2,294.0 | 7 | 8 | 100.0% | 2,433.4 |
| 1 | 84 | 100.0% | 2,227.6 | 1 | 52 | 67.3% | 2,239.3 | 3 | 23 | 100.0% | 2,311.1 |
| 5 | 36 | 2.8% | 2,207.2 | 5 | 38 | 0.0% | 2,213.4 | 4 | 52 | 7.7% | 2,133.8 |
| 0 | 33 | 0.0% | 2,161.5 | 9 | 25 | 0.0% | 2,209.8 | 1 | 80 | 57.5% | 2,065.2 |
| 2 | 36 | 0.0% | 2,155.5 | 6 | 25 | 12.0% | 2,207.6 | 5 | 78 | 20.5% | 2,030.8 |
| 4 | 60 | 0.0% | 2,078.2 | 2 | 37 | 0.0% | 2,192.4 | 6 | 96 | 41.7% | 1,932.1 |

## C   MODEL AGNOSTICISM AND POTENTIAL GENERALIZATION

In this section, we clarify the relationship between our geometric framework and the Factor Analysis (FA) model. The agnosticism of the geometric lifting stems from its reliance solely on the residual field $r$, making the framework applicable to any generative architecture. Whether the reconstruction originates from a linear factor model, a Variational Autoencoder (VAE), or a Diffusion Model, the lifting process—mapping local residual matrices $E_i$ to the Grassmannian $Gr(m, D)$—remains mathematically identical.

However, the diagnostic power of our metric requires a reference to distinguish "meaningful" geometric structures from diffuse stochasticity. Factor Analysis serves as this baseline because its probabilistic formulation explicitly partitions variance into common factors and independent stochastic noise ($\Psi$). This explicit noise model provides the necessary contrast to identify structural deviations via $D_{KL}$.

For non-linear models where the noise distribution is often implicit, this framework can be generalized by estimating a reference $\Psi$ empirically. By calculating the reconstruction error covariance on a balanced validation set, one can establish a baseline noise floor. Thus, the $D_{KL}$ test could remain a viable diagnostic for structural bias even in deep latent-variable models, provided that a representative stochastic baseline is estimated.

## D  Hyperparameter and Thresholding Selection

In this section, we provide the rationale and empirical justification for the hyperparameters used in our diagnostic framework, specifically addressing the thresholds for the Kullback-Leibler divergence and the clustering parameters.

### KL-Divergence

The $D_{KL}$ serves as a measure of structural "leakage" into the residual field. To determine a reasonable threshold for flagging systematic failure, we established a baseline noise floor using a balanced model. In our control experiments, residuals consistent with stochastic noise yielded $D_{KL} \approx 2.4 \times 10^3$. In contrast, biased scenarios exhibited $D_{KL}$ values jumping to $\approx 2.7 \times 10^5$. Our threshold is based on orders of magnitude. A cluster is flagged as "Structural" if its $D_{KL}$ is at least one order of magnitude higher than the baseline noise covariance $\mathbf{\Psi}$. This heuristic ensures that we do not flag minor sampling fluctuations, but only significant deviations.

### Clustering Thresholds ($K$):

For the Spectral Clustering on the Grassmann manifold, we set $K = 10$. Although our experiments focused on a binary subset of MNIST (digits 1 and 7), we maintained $K > 2$ to allow for over-segmentation of the failure modes. Setting $K = 10$ ensures that the algorithm is not merely recovering the digit labels, but is instead forced to isolate fine-grained morphological variants of failure within each class—such as distinguishing between errors caused by extreme slant versus those caused by unusual stroke thickness.

