# OpenReview forum: "Towards a Geometric Theory of Fairness: Detecting Mode Collapse on the Grassmannian Manifold"
_ICLR.cc/2026/Workshop/GRaM — ICLR 2026 Workshop GRaM Poster_

### Official Review · Reviewer_N8xP · 2026-02-08

**Rating:** 6
**Confidence:** 3

**Review:**

A geometric viewpoint of fairness is introduced, using the idea of subspace stability. They argue that systematic unfairness cannot be modeled via stochasticity (diffusion), and they propose a geometric mode collapse viewpoint via low-dimensional subspaces. The canonical model for subspaces in geometry is the Grassmannian, and the authors use it to build their framework. They present early experiments on MNIST to show that their manifold perspective on the problem is relevant, and it beats the baseline Euclidean approaches.


pros:
- geometric viewpoint into a problem (definitely relevant to the workshop)
- nice formulation and novel results

cons:
- early experiments (ok for workshop version though)

This is an inteteresting paper, and for a workshop it is ok to have proof of concept experiements but please make sure to extend your experiments for the next version

**Pmlr Suitability:**

NA

---

### Official Review · Reviewer_r4ck · 2026-02-10
**Interesting Preliminary Paper**

**Rating:** 6
**Confidence:** 4

**Review:**

## Methodology Overview
This paper addresses the problem of distinguishing random errors from genuine model failures for subpopulations of the data by providing a tool to discriminate between and group such cases. The argument put forward can be summarised as follows: genuine failures of the model will have a residual tangent subspace that tends to align across misclassified clusters, whereas noise does not exhibit any geometric tendency; instead, it is 'diffuse' in structure. The discrimination is at first achieved through the use of the Kullback-Leibler divergence, between the sample’s residual divergence and a baseline noise modelled as a multivariate normal derived from the Factor Analysis model.

To assess whether a set of residuals is structurally similar, the nearest-neighbour samples surrounding a high-reconstruction error sample are grouped into a matrix, and the subspace spanned by the top-m singular vectors is computed from this matrix. The argument is that noise will lead to randomly oriented tangent subspaces, whereas systematic underperformance on a subpopulation will lead to several local clusters of residual subspaces with a consistent directionality.

These are then grouped by pairwise comparison of orthonormal bases spanning each cluster’s residual tangent subspace.

## Quality and Clarity
The problem is well-motivated, with a justified geometric methodology for distinguishing noise from structural model failures, given the assumptive premises (e.g., the nature of the noise, the choice of generative model, etc.). The mathematics is clear and concisely provided. For clarity, the manuscript was well-written, but due to the length limit, it necessitated only brief expositions of concepts, but remains well-suited to the existing geometrical background of the GRaM audience. The following points raised are suggestions to further improve quality and clarity.

The work would benefit from an appendix listing all the notation for readers' reference. This will add greater clarity for readers. Additionally, aspects of the methodology were underspecified, namely the thresholding (and justification therein) of the KL-divergence for randomness versus structure. Similarly, for the thresholding for clustering. Discussion of how reasonable values for these were determined would add clarity.

At this preliminary stage, it appears that the approach is limited to linear Factor Analysis as the generative model. The KL-divergence approach is non-trivial to generalise to other models due to non-linearities, thereby forfeiting the reference normal distribution used to distinguish structure from randomness, available in the Factor Analysis model. Explicit upfront stating of this Factor analysis dependence would be of benefit to readers, and an appendix discussing possible generalisations of the methodology outside of Factor Analysis, alongside a more detailed discussion of its existing limitations. For example, to what extent does the methodology presuppose factor analysis as a model?

The empirics in support of the approach are very limited, demonstrated on solely a modified MNIST dataset, and several are qualitative and largely open to interpretation. This does not indicate how the model generalises to other datasets, including further standard benchmarks which offer more complexity than MNIST, e.g. even the toy-dataset CIFAR could have almost identical methodology applied and would have greatly enriched reader confidence in the method. It is strongly recommended that, if accepted, additional appendices containing other datasets be included to provide readers with a clearer understanding of how the methodology generalises beyond very simple datasets.

Further, the qualitative descriptions are ambiguous: “distinct horizontal strokes and acute angles” - personally, I measured substantial overlap across all six samples in terms of angles: approximately 53, 70, 71, 47, 72 and 68 degrees for left to right, first row, then second, respectively, for figure 1. Other quantitative empirical findings are reasonable and indicate that the approach is successful.

Minor point: Outside of a GRaM-only audience, I do not feel the primary narrative necessitates the formal Grassmannian framing, despite being geometrically accurate. The tool remains geometrically intuitive without this framing and may have broader applicability if this formality is provided as background auxiliary information. Absent is leveraging the Grassmannian geometry for proofs of performance, theoretical guarantees on convergence, etc., as this is reasonable for a tiny-paper track and preliminary methodology. However, unless it is felt that this formality conveys some methodological superiority heuristically to assure readers, or the geometry is of primary interest, perhaps its centrality should be reduced. I state this because the methodology does not make heavy use of such geometric structure, aside from the residual principle subspace happening to be an element of the Gr(m, D) manifold and the basis-independent metric for clustering. This is necessarily subjective, but the framing was felt to be rhetorically overemphasised compared to its use, and this foregrounding of Grassmannian geometry may be revised in future works, with the advantage of also attracting a wider readership, by ensuring the core algorithmic steps do not invoke this full cognitive load to the degree that the framing suggests.

## Originality and significance
The work clearly positions itself against the existing literature, particularly in Appendix A, which presents direct comparisons with Stiefel and Euclidean baselines. The practical use of Grassmannian geometry as a foundation appears original to this approach and the motivation for its use is provided. The connection of this geometric structure may provide greater context to readers of GRaM.

In terms of significance, this is an early work empirically but provides a sound conceptual and methodological contribution. However, the empirical evidence is very limited, having been demonstrated only on MNIST. However, the limited empirical evidence does not significantly detract from the overall methodological contribution of this early work. Hence, it is promising and offers a conceptual contribution, but remains empirically incomplete. Results obtained across a wider range of datasets are recommended for publication.
### Pros:
- Conceptually well-motivated problem which introduces novel, straightforward tooling.
- Anchors this through geometric principles, which is appropriate for the GRaM audience.
- Performs ablation comparison against Euclidean and Stiefel approaches in Appendix A.
- Modifies the MNIST dataset so as to compare performance on a heavily biased dataset compared to the baseline.
### Cons:
- Some methodological details are underspecified, namely the divergence threshold used and the threshold for projection distance.
- Is a preliminary methodology and appears contingent on Factor Analysis, which is not clear upfront.
- Very limited empirical evidence, limited to toy-MNIST examples with qualitative interpretation of samples.
- Lacking a notation table as a reference
- The strong geometric framing through Grassmannian geometry is accurate, but as a primary motivation, it is unnecessary for a wider audience. It does not leverage the deeper structure of this specific geometry beyond a relation to PCA and the metric; it isn’t used to provide proofs of performance and generally serves as an interesting connection and formalism for heuristic assurance, but may dissuade a broader audience.

**Pmlr Suitability:**

NA

---

### Official Review · Reviewer_UXqr · 2026-02-22

**Rating:** 6
**Confidence:** 3

**Review:**

## Summary

This paper proposes a geometric framework for fairness auditing in generative models. **The central hypothesis is that systematic unfairness manifests as stable, low-rank subspaces in residual space rather than large Euclidean errors.** The authors lift local residual matrices onto the Grassmann manifold and cluster the resulting subspaces using a projection kernel. A KL divergence test based on a Factor Analysis noise model is used to distinguish structural collapse from stochastic noise.

Experiments on an imbalanced MNIST setting show that the Grassmannian approach consolidates minority failure modes into coherent clusters and produces significantly elevated KL divergence under bias, while remaining stable in balanced scenarios. The work aligns most strongly with *Geometry in theoretical analysis* and *Data and latent geometry* within the GRaM topics.

------

## Strengths

- Appropriate use of Grassmann geometry to ensure rotational invariance of subspaces.
- Combination of geometric clustering with a KL-based statistical diagnostic adds interpretability.
- Controlled balanced vs. biased experiments provide proof-of-concept validation.

------

## Weaknesses

- Experimental validation is limited to a simplified MNIST imbalance scenario.
- Some methodological details (e.g., notation and dimensionalities in Section 2.1) are under-explained.
- Sensitivity to hyperparameters (rank, neighborhood size, number of clusters) is not analyzed.

**Pmlr Suitability:**

NA

---

### Meta-Review · Area_Chair_N3Cm · 2026-02-26

**Decision:**

Accept

**Metareview:**

The paper shows that bias can be detected (without labels), and for this, they look at low-dimensional subspaces, linked to Grassmanian geometry. This a well-motivated and novel paper. We are happy to accept it for our GRaM workshop.

**Relevance To Proceedings:**

Tiny paper — does not apply

**Relevance To Workshop:**

Yes — suitable for GRaM

---

### Decision · Program_Chairs · 2026-03-02

Accept (Poster)